# Serum Anti-Müllerian Hormone in the Diagnosis of Polycystic Ovary Syndrome in Association with Clinical Symptoms

**DOI:** 10.3390/diagnostics9040136

**Published:** 2019-10-01

**Authors:** Nada Ahmed, Asma A. Batarfi, Osama S. Bajouh, Sherin Bakhashab

**Affiliations:** 1Biochemistry Department, King Abdulaziz University, P.O. Box 80218, Jeddah 21589, Saudi Arabia; nahmed0028@stu.kau.edu.sa (N.A.); asma.batarfi@hotmail.com (A.A.B.); 2Department of Obstetrics and Gynecology, Faculty of Medicine, King Abdulaziz University, P.O. Box 80205, Jeddah 21589, Saudi Arabia; dr_bajouh@yahoo.com; 3Center of Innovation in Personalized Medicine, King Abdulaziz University, P.O. Box 80216, Jeddah 21589, Saudi Arabia

**Keywords:** polycystic ovary syndrome, anti-Müllerian hormone, oligo/amenorrhea, hyperandrogenism

## Abstract

Polycystic ovary syndrome (PCOS) is one of the most prevalent endocrine diseases affecting women of reproductive age. The pathogeny of PCOS is still not completely understood, but one contributing factor that has been proposed is anti-Müllerian hormone (AMH). There is currently no clear correlation between levels of AMH and incidence of PCOS in Saudi Arabian patients. The goal of this study was to determine the threshold of AMH and correlate it with PCOS clinical features to facilitate a proper diagnosis for PCOS. In this case-control study, we recruited 79 PCOS women and 69 normal ovulatory women; PCOS patients were diagnosed according to the Rotterdam criterion. On days 2–4 of the menstrual cycle, transvaginal/abdominal ultrasound was performed and serum levels of AMH, luteinizing hormone (LH), and follicle-stimulating hormone (FSH) were measured for all participants. The receiver operating characteristic curve (ROC) was used to determine the AMH diagnostic cut-off at 3.19 ng/mL, with 72% sensitivity and 70% specificity; AMH > 3.19 ng/mL was significantly correlated with PCOS. High AMH levels were correlated with age at menarche, polycystic ovarian morphology (PCOM), and oligo/amenorrhea. Serum AMH is a promising diagnostic marker of ovarian dysfunction in PCOS patients especially in cases in which the evaluation of PCOM was complicated.

## 1. Introduction

Polycystic ovary syndrome (PCOS), a multifactorial syndrome, is one of the most common endocrine diseases affecting reproductive-aged women, with an incidence rate of 9–18% [1]. Currently, it is diagnosed according to the Rotterdam criterion that defines PCOS by the presence of two out of three of the following symptoms: polycystic ovarian morphology (PCOM), clinical or biochemical hyperandrogenism (HA) and oligo/amenorrhea (OA) [2,3]. PCOM is diagnosed by the presence of at least 12 follicles 2–9 mm in diameter, and measured using transvaginal or abdominal ultrasound [2]. Ethnic and racial differences in follicle number per ovary and ovarian volume in determining PCOM diagnostic cut-offs have been measured in Chinese and Turkish women [4,5].

Recent research concerning different aspects of PCOS has produced new insights into assessment and treatment of the disorder. The relationship between PCOS and anti-Müllerian hormone (AMH) is of interest as AMH plays an important role in ovarian function. It preserves the follicles in the primordial stage and estimates the number of ova in the ovaries, thereby providing an indication of the ovarian reserve [6,7]. Previous studies have shown that serum AMH levels correspond with the number of antral follicles which are augmented in PCOS and PCOM [8,9]. Women with PCOS were found to have higher serum AMH due to a greater number of antral follicles and greater production of AMH per follicle [10,11], making AMH possibly an important factor in the diagnosis of PCOS. Dewailly et al. [12] proposed AMH levels greater than 5 ng/mL could be a diagnostic substitution for PCOM, while another study suggested AMH levels greater than 3.8 ng/mL [13]. The presence of two out of three clinical features (OA, HA, and/or AMH) was found to have 96% sensitivity and 100% specificity among patients previously diagnosed with PCOS, according to the Rotterdam criterion [13]. Moreover, Dewailly et al. [14] developed merged PCOS diagnosis criteria using Rotterdam and non-Rotterdam definitions, proposing an excessive follicle number or serum AMH concentration as a surrogate marker for the diagnosis of PCOS in the absence of either OA or HA [14]. Previous studies have shown the potency of serum AMH as a diagnostic marker for PCOS and in determining the optimum diagnostic level [9,12]. Furthermore, a pregnant mouse model treated with high AMH during pregnancy caused changes in the fetus, which ended up having a PCOS-like disorder in adulthood [15]. Ethnicity has been associated with variable AMH levels, and Asian women have lower AMH at a given age than Caucasians [16]. The AMH diagnostic cut-off has been demonstrated at 4.7–5 ng/mL in Caucasians [17], and 10 ng/mL in Japanese and Korean women [18,19]. In this study, we aim to determine the PCOS diagnostic cut-off for AMH in the Saudi population. We also study the relationship between the serum AMH threshold in patients and prominent clinical parameters of PCOS. This will allow for assessment of the diagnostic accuracy of using AMH as a surrogate compared to the Rotterdam criterion, and may contribute to the proper diagnosis of PCOS among Saudi women.

## 2. Materials and Methods

### 2.1. Study Design

This was a case-control, observational, and randomized study implemented in the Obstetrics and Gynecology Clinics, King Abdulaziz University Hospital, Jeddah, Saudi Arabia and Center of Innovation in Personalized Medicine (CIPM), KAU, Jeddah, Saudi Arabia, between 2016–2018. The clinical study, sample size selection, and exclusion criteria have been previously discussed [20]. Using Raosoft (www.raosoft.com), the appropriate sample size was calculated to be 196 women of reproductive age between 18 and 38 years old. The study was approved by the Biomedical Ethics Unit, Faculty of Medicine, KAU (approval number: 407–15) on 19 January 2016 and written informed consent was obtained from participants prior to sample collection. The study was conducted in accordance with the Declaration of Helsinki.

The case group was composed of 98 selected PCOS patients diagnosed based on the Rotterdam Criteria Consensus, and the control group was composed of 98 women with normal ovulation. The enrolled women were not under medication. On days 2–4 of the menstrual cycle, transvaginal or abdominal ultrasound (SonixTouch, Ultrasonix Medical Corporation, Richmond, BC, Canada) were performed, and serum hormonal levels for luteinizing hormone (LH) and follicle-stimulating hormone (FSH) were measured with an automated multi-analysis system using electrochemiluminescence immunoassay kits (Roche, Basel, Switzerland). Serum AMH levels were determined by an enzyme-linked immunosorbent assay using an Ultra-Sensitive Anti-Müllerian hormone/Müllerian inhibiting substance Kit (AnshLabs, Webster, TX, USA) according to the manufacturer’s instructions.

HA was assessed clinically through the presence of hirsutism, androgenic alopecia, and acne. Patients’ hirsutism was evaluated by a Ferriman–Gallwey score greater than 8. OA was defined as less than eight menstrual cycles in 12 months, or if the menstrual interval was more than 35 days. Using transvaginal or abdominal ultrasound, PCOM was identified by the presence of at least 12 follicles measuring 2–9 mm in diameter in one ovary [2]. Poor quality samples were excluded from the statistical analyses, resulting in 79 cases and 69 controls.

Initially, patients were diagnosed according to the Rotterdam criterion. After the selection of the AMH threshold value at 3.19 ng/mL, the PCOS group was re-evaluated twice. We used two different classification methods. The first classification identified PCOS as exhibiting two out of three features by which AMH may replace PCOM: OA, HA, or AMH > 3.19. The second classification defined PCOS s having two out PCOM, HA, or AMH > 3.19. After that, two new receiver operating characteristic (ROC) curves (one for each classification) were constructed to assess the sensitivity and specificity of a 3.19 ng/mL AMH diagnostic cut-off.

### 2.2. Statistical Analysis

A normality test was performed to determine if the data were normally distributed. Data are presented as (median ± interquartile range (IQR)), *p*-values were calculated using a Mann-Whitney test. The ROC curve was utilized to examine the diagnostic cut-off of AMH. The chi-squared test was used to demonstrate the correlation between AMH, PCOS, and PCOS variables. In the PCOS group, the association was evaluated using binary logistic regression or multinomial logistic regression between AMH cut-off and PCOS clinical features. Statistical analyses were performed using IBM SPSS software version 24.0 (SPSS Inc., Armonk, NY, USA). A *p*-value < 0.05 was considered statistically significant.

## 3. Results

Clinical characteristics are summarized in Table 1. The case and control groups were matched for age (23 ± 9 vs 21 ± 3.5, *p* = 0.067) but not on body mass index (BMI, 25.1 ± 6.98 kg/m^2^ vs 23 ± 5.94 kg/m^2^, *p* = 0.005). The serum levels of AMH and LH were significantly higher in the case group than in the controls (4.8 ± 4.76 ng/mL vs 2.3 ± 1.41 ng/mL, *p* < 0.0001; and 8.4 ± 8.55 IU/mL vs 5.8 ± 5.2 IU/mL, *p* = 0.003, respectively). Although there was no significant difference in serum levels of FSH, the LH/FSH ratio was significantly higher amongst the case group compared to the control group (1.7 ± 1.79 vs 1.2 ± 1.47, *p* = 0.006). In the PCOS group, 73 women had PCOM (92%), 66 women had AO (83%), and 55 women had symptoms of HA (69.6%).

To determine the AMH diagnostic cut-off, the ROC curve was applied (Figure 1). A 3.19 ng/mL AMH cut-off level was determined with 72% sensitivity, 70% specificity, and the area under the curve equaling 0.785 (0.713–0.858), with a 95% confidence interval. Greater than 3.19 ng/mL was determined to be significantly associated with PCOS (*p* < 0.001) using the chi-squared test.

Pearson’s correlation analysis revealed a positive correlation between AMH and age at menarche (*r* = 0.324, *p* = 0.018) and no correlation between AMH and BMI, LH, FSH, and LH/FSH ratio (Table 2).

Additionally, a chi-squared test was performed on the clinical features of PCOS and AMH cut-off levels in the case group. There was a significant association (Table 3) between AMH at the threshold cut-off with two variables, PCOM (*p* = 0.048) and OA (*p* = 0.015), whereas no association was detected with HA (*p* = 1.000).

Logistic regression analysis performed between AMH ≥ 3.19 and PCOM or OA revealed a significant correlation with PCOM (odds ratio (OR) = 6.1, B = 1.8, *p* = 0.046) and no correlation with OA. Moreover, multinomial logistic regression also showed a positive correlation between AMH ≥ 3.19 and age at menarche (OR = 1.8, B = 0.571, *p* = 0.030). In other words, women with PCOS who experienced menarche at an older age were 1.8 times more likely to have higher AMH.

After re-assessing the case group using the AMH ≥ 3.19 cut-off, the PCOM was substituted with AMH (first classification) to give the following distribution of patients: 73 still PCOS, and 6 non-PCOS. The AMH cut-off was re-calculated, and a sensitivity of 78% and specificity of 100% were obtained at AMH = 3.19 ng/mL (Figure 2). When AMH substituted OA, the PCOS group included 70 patients, while the non-PCOS group included 9. An AMH cut-off greater than 3.19 was found to have 81% sensitivity and 100% specificity (Figure 3).

## 4. Discussion

The etiology of PCOS is still unclear and the ability to make an integrated diagnosis among clinicians is difficult due to subjective phenotypes. Furthermore, it is well established that PCOS is associated with insulin resistance. If this fact is combined with delayed or confused PCOS diagnosis, then the prognosis of type 2 diabetes and cardiovascular disease may be accelerated [21,22,23]. Moreover, the evaluation of PCOM through abdominal ultrasound can be difficult, especially in virgins or obese women. Thus, easier, clearer, and more quantitative diagnostic criteria are essential. Serum AMH can now function within the diagnostic criteria, as serum AMH level has not only shown a significant association with PCOS, but can also reflect the severity of the disease [24,25]. Moreover, it can be easily measured at any time during a woman’s cycle [26].

In this study of a sample among the Saudi population, we assessed whether serum AMH was a possible additional tool for the diagnosis of PCOS regardless of patient phenotype. Our results showed a significant difference in AMH levels among the control and PCOS groups. The latter demonstrated a two-fold increase in AMH over controls, similar to those found in previous studies [27,28].

It has been suggested that AMH may be a useful diagnostic test for PCOS with cut-off thresholds ranging from >2.8 to >8.16 ng/mL [29,30]. As in prior studies, we found that AMH > 3.19 ng/mL was significantly associated with PCOS. The variations in AMH threshold may be attributed to many factors including ethnicity and analytical method variations [31]. The existence of such racial/ethnic differences in ovarian reserves, as reflected by AMH, may be also attributed to various genetic factors and environmental factors such as obesity, smoking, and vitamin D deficiency, which have been shown to correlate with serum AMH levels [32].

Menarche age showed a significant association with AMH, as we found that PCOS patients experiencing menarche at an older age were more prone to have high AMH levels. While previous studies have concluded that women who experience early menarche have significantly higher AMH as young adults [33,34], this was contrary to our findings. This could be attributed to the variation in the tested populations, as both studies were conducted among non-PCOS subjects.

Moreover, our results showed that women with PCOS with AMH > 3.19 ng/ mL had a greater prevalence of PCOM and OA. A previous study proposed that women with AMH > 10 ng/mL have a greater prevalence of PCOS and OA [35]. Additionally, we have demonstrated that there is no association between AMH cut-off and HA, while others have suggested AMH as a surrogate marker for HA [14].

The accuracy of AMH as a diagnostic criterion was previously examined in two groups by re-assessing the diagnosis of PCOS using Rotterdam criteria. They found that when PCOM in Rotterdam criteria was replaced by AMH, any two of AMH, OA or HA provided a sensitivity of 86.67% [28] or 83% [14]. After applying this classification to our cohort we achieved 78% sensitivity. Alternatively, when AMH was replaced by OA, 100% specificity and 81% sensitivity were obtained. Our results are similar to those in a previous study (92% specificity and 79% sensitivity) [35]. Serum AMH levels in another recent study added values in classifying different PCOS phenotypes that might help clinicians identify patients at high risk of ovarian hyperstimulation syndrome and customize specific therapies [36].

## 5. Conclusions

There is a positive correlation between high serum AMH levels, PCOM, OA, age of menarche, and PCOS. AMH is a key diagnostic marker of ovarian dysfunction in PCOS patients in combination with other clinical features, especially in cases with ambiguous evaluation of PCOM via ultrasound. Although there is no single diagnostic cut-off for AMH, it is still an objective and quantitative marker not affected by the day of menses. Future studies in different regions of Saudi Arabia should be undertaken to validate the cut-off.

## Figures and Tables

**Figure 1 diagnostics-09-00136-f001:**
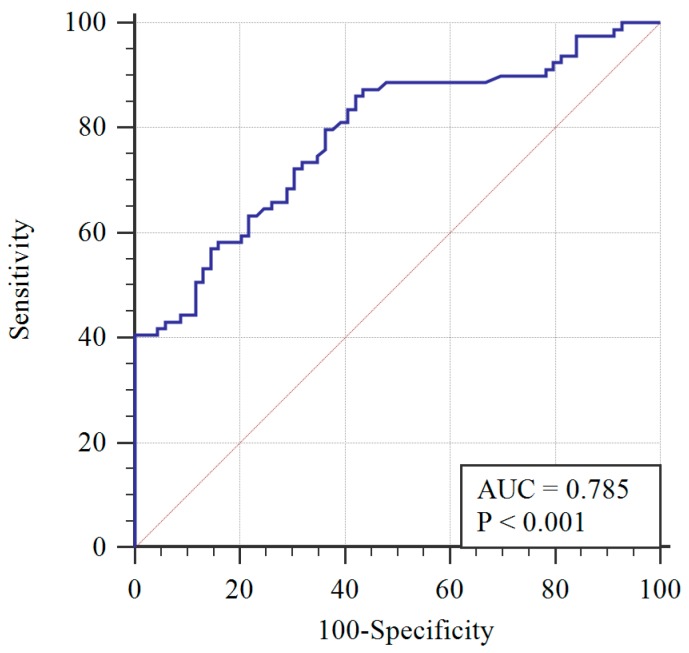
Receiver operating characteristic (ROC) curve of AMH with respect to PCOS diagnosis (blue line) and reference curve (red line). AUC: Area under the curve.

**Figure 2 diagnostics-09-00136-f002:**
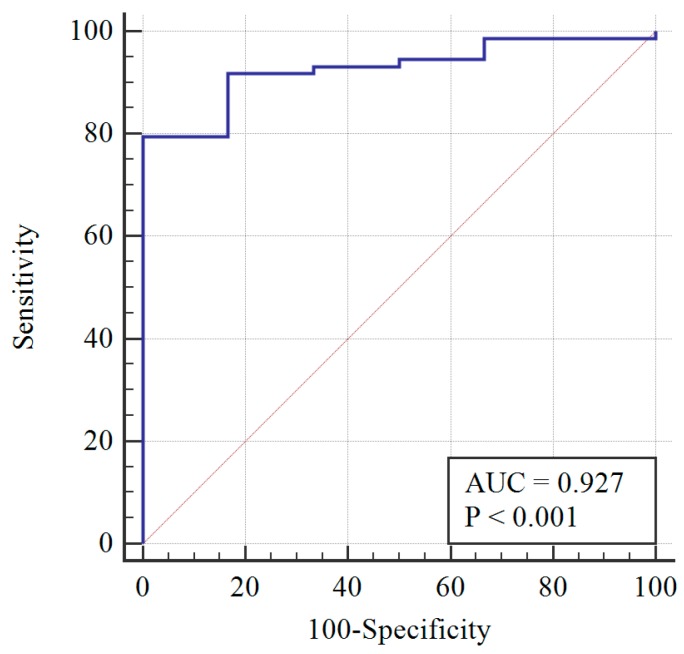
Diagnosis capability of OA, HA, or AMH > 3.19 ng/mL across the PCOS group (blue line) and reference curve (red line). AUC: Area under the curve.

**Figure 3 diagnostics-09-00136-f003:**
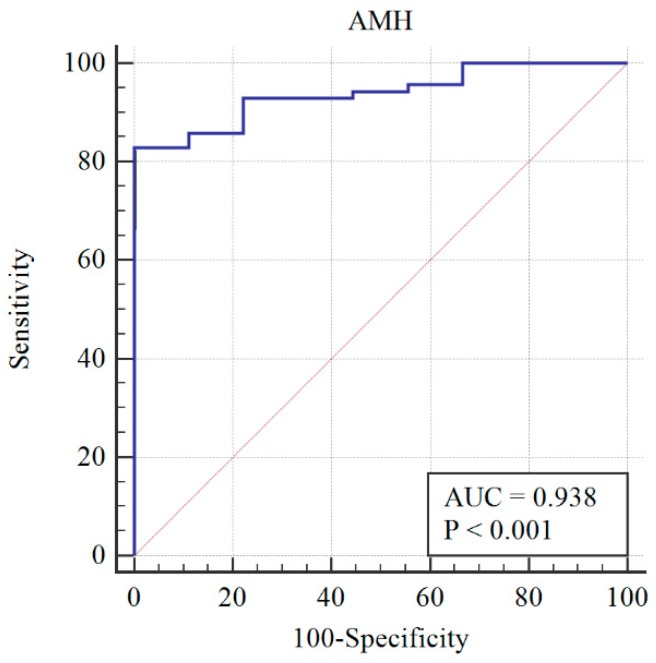
Diagnosis capability of PCOM, HA, or AMH > 3.19 ng/mL across the PCOS group (blue line) and reference curve (red line). AUC: Area under the curve.

**Table 1 diagnostics-09-00136-t001:** A summary of the clinical characteristics of polycystic ovary syndrome (PCOS) patients and control subjects.

Variable	Control (*n* = 69)	PCOS (*n* = 79)	*p*-Value
Age (years)	21 ± 3.5	23 ± 9	0.067
BMI (kg/m^2^)	23 ± 5.94	25.1 ± 6.98	0.005 **
LH (IU/L)	5.8 ± 5.2	8.4 ± 8.55	0.003 **
FSH (IU/L)	4.6 ± 2.75	4.7 ± 2.45	0.618
LH/FSH ratio	1.2 ± 1.47	1.7 ± 1.79	0.006 **
AMH (ng/mL)	2.3 ± 1.41	4.8 ± 4.76	<0.0001 ***

Values are expressed as median ± interquartile range (IQR), and *p*-values were calculated using a Mann-Whitney test for data with non-normal distribution. *p*-value < 0.05 is statistically significant. BMI: body mass index; LH: luteinizing hormone; FSH: follicle-stimulating hormone; AMH: anti-Müllerian hormone. ** *p* < 0.01, *** *p* < 0.001.

**Table 2 diagnostics-09-00136-t002:** Associations between AMH cut-off and PCOS scale factors.

Clinical Characteristics	*r*	*p*-Value
Age (years)	−0.113	0.328
Age at menarche (years)	0.324	0.018 *
BMI (kg/m^2^)	−0.037	0.750
LH (IU/L)	0.031	0.794
FSH (IU/L)	0.019	0.870
LH/FSH ratio	0.073	0.540

*p*-values were calculated by Pearson’s correlation test and Pearson’s correlation coefficient *r* is used to present the associations. *p*-values < 0.05 are statistically significant. BMI: Body mass index; LH: luteinizing hormone; FSH: Follicle-stimulating hormone. * *p* < 0.05.

**Table 3 diagnostics-09-00136-t003:** Association between PCOS clinical symptoms and AMH at threshold cut-off level.

Clinical Symptoms	AMH ≥ 3.19	AMH < 3.19	*p*-Value
PCOM (Yes/No)	55/2	18/4	0.048
OA (Yes/No)	44/13	22/0	0.015
HA (Yes/No)	39/13	16/6	1.000

*p*-values were calculated by a chi-squared test. *p*-values < 0.05 are statistically significant. PCOM: Polycystic ovarian morphology; OA: Oligo/amenorrhea is defined by less than eight periods per year or a period length > 35 days; HA: Hyperandrogenism is defined by the presence of hirsutism, acne, or androgenic alopecia.

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
