# Peer review of "Serum Anti-Müllerian Hormone in the Diagnosis of Polycystic Ovary Syndrome in Association with Clinical Symptoms"

_diagnostics, 2019, doi:10.3390/diagnostics9040136_

Round 1
Reviewer 1 Report
I read with great interest the Manuscript titled “Serum Anti-Müllerian Hormone as a Diagnostic Tool for Polycystic Ovary Syndrome in Association with Clinical Symptoms” (diagnostics-600850), which falls within the aim of Diagnostics. The Authors performed a case-control study aimed to determine the threshold of AMH and correlate it with PCOS clinical features to facilitate the proper diagnosis of PCOS. In my honest opinion, the topic is interesting enough to attract the readers’ attention. Nevertheless, authors should clarify some methodological points and improve the discussion citing relevant and novel key articles about the topic.
Authors should consider the following recommendations:
Manuscript should be further revised by a native English speaker. I would suggest improving the description of the study population reporting which inclusion and exclusion criteria were used (age, hormonal therapy,) and from where (ambulatory, general population) the patients come from. The Authors did not mention the sample size calculation for their study. It is essential to specify this data in order to guarantee an adequate significance of the results obtained by the Authors. I would suggest better clarifying the three steps of the study in the methods section. The authors have not adequately highlighted the strengths and limitations of their study. I suggest better specifying these points. It is mandatory to discuss that to confirm the study results and the value of AMH for the diagnosis of PCOS, the application of the proposed classifications on another population is required. What are the actual clinical implications of this study? it is important to report the results obtained by the authors in the context of clinical practice and to adequately highlight what contribution this study adds to the literature already existing on the topic and to future study perspectives. Accumulating evidence suggests that one of the most important mechanisms of PCOS pathogenesis is the insulin-resistance. For this reason, the use of insulin-sensitizers, such as inositol isoforms, gained increasing attention due to their safety profile and effectiveness. Authors may better discuss this point, taking to account these recent articles: PMID: 28835764; PMID: 26177098. As reported by the Authors, high AMH level is a characteristic of PCOS. Nevertheless, in these patients AMH seems to lose its usual role as marker of ovarian reserve. In example, to many efforts are spent to identify a correct algorithm which consider woman's age and ovarian reserve markers (AMH) as a tool to optimize the follicle-stimulating hormone starting dose in IVF procedure. Nevertheless, current available evidence regarding PCOS women, particularly the ones with high AMH, suggest that AMH does not seem adequate for this interpretation. I would be glad if the authors discuss this important fact, referring to: PMID: 30311070; PMID: 27835829.Author Response
Comments and Suggestions for Authors
I read with great interest the Manuscript titled “Serum Anti-Müllerian Hormone as a Diagnostic Tool for Polycystic Ovary Syndrome in Association with Clinical Symptoms” (diagnostics-600850), which falls within the aim of Diagnostics. The Authors performed a case-control study aimed to determine the threshold of AMH and correlate it with PCOS clinical features to facilitate the proper diagnosis of PCOS. In my honest opinion, the topic is interesting enough to attract the readers’ attention. Nevertheless, authors should clarify some methodological points and improve the discussion citing relevant and novel key articles about the topic.
Point 1:Manuscript should be further revised by a native English speaker.
Response 1:Thank you the manuscript had been revised by Cambridge proofreading and editing LLC.
Point 2:I would suggest improving the description of the study population reporting which inclusion and exclusion criteria were used (age, hormonal therapy,) and from where (ambulatory, general population) the patients come from.
Response 2:Thank you for this comment. The exclusion criteria:
Condition with reproductive symptoms similar to PCOS, including congenital adrenal hyperplasia, Cushing syndrome, hyperprolactinemia, thyroid disease, and androgen-secreting tumors. Chronic diseases, including diabetes, cardiovascular disease. Any other female infertility factor.The exclusion criteria were referred to in Methods section- Study Design, in the reference Batarfi, A. A., et al. (2019). "MC4R variants rs12970134 and rs17782313 are associated with obese polycystic ovary syndrome patients in the Western region of Saudi Arabia."BMC Med Genet20(1): 144.
The enrolled women were not under medication.
Point 3: The Authors did not mention the sample size calculation for their study. It is essential to specify this data in order to guarantee an adequate significance of the results obtained by the Authors. I would suggest better clarifying the three steps of the study in the methods section.
Response 3: The sample size selection was discussed in detail in the reference Batarfi et.al, 2019and summarized in the ‘Study Design’ section.
Point 4:The authors have not adequately highlighted the strengths and limitations of their study. I suggest better specifying these points. It is mandatory to discuss that to confirm the study results and the value of AMH for the diagnosis of PCOS, the application of the proposed classifications on another population is required.
Response 4:Thank you for this comment. The strengths and limitations of the study were added in the conclusions section, lines 218-220.
Point 5:What are the actual clinical implications of this study? it is important to report the results obtained by the authors in the context of clinical practice and to adequately highlight what contribution this study adds to the literature already existing on the topic and to future study perspectives.
Response 5:Thank you for this comment. This had been discussed in the ‘discussion’ section, lines 188-192.
Point 6:Accumulating evidence suggests that one of the most important mechanisms of PCOS pathogenesis is the insulin-resistance. For this reason, the use of insulin-sensitizers, such as inositol isoforms, gained increasing attention due to their safety profile and effectiveness. Authors may better discuss this point, taking to account these recent articles: PMID: 28835764; PMID: 26177098.
Response 6: Thank you for this comment. The enrolled women were not under medication, we collected the samples before medication.
Point 7: As reported by the Authors, high AMH level is a characteristic of PCOS. Nevertheless, in these patients AMH seems to lose its usual role as marker of ovarian reserve. In example, to many efforts are spent to identify a correct algorithm which consider woman's age and ovarian reserve markers (AMH) as a tool to optimize the follicle-stimulating hormone starting dose in IVF procedure. Nevertheless, current available evidence regarding PCOS women, particularly the ones with high AMH, suggest that AMH does not seem adequate for this interpretation. I would be glad if the authors discuss this important fact, referring to: PMID: 30311070; PMID: 27835829.
Response 7:Thank you for your comment. In a recent study, serum AMH levels added values in classifying different PCOS phenotypes in order to compare the ovarian response to controlled ovary stimulation and IVF outcome (Cela, V., et al. (2018). "Ovarian response to controlled ovarian stimulation in women with different polycystic ovary syndrome phenotypes." Gynecol Endocrinol34(6): 518-523. I checked the two suggested articles, and I found them irrelevant as PMID: 30311070 is on drug metabolism in PCOS and PMID: 27835829 on clinical application of a nomogram based on age, serum FSH and AMH to select the starting dose of FSH in IVF cycles but it was conducted on non-PCOS patients.
This point had been discussed in the manuscript lines 213-215.
Reviewer 2 Report
This study looks at the use of AMH concentration of AMH as a surrogate marker of PCOSPCOS in Saudi Arabia women population.
Methods, results and interpretation of data are clear.
However introduction and discussion need more explanation about the relevance of ethnicity and the racial differences of AMH cutoff value for the diagnosis of PCOS with possible interpretation.
Since the authors mention in the title " association with clinical symptoms", there should more analysis of this data, in particular regarding the AMHlevels in various phenotypes of PCOS. I suggest to specify the evaluation of hirsutism with Ferriman-Gallwey score.
Author Response
Comments and Suggestions for Authors
Point 1:This study looks at the use of AMH concentration of AMH as a surrogate marker of PCOS in Saudi Arabia women population.
Methods, results and interpretation of data are clear.
Response 1: Thank you.
Point 2:However, introduction and discussion need more explanation about the relevance of ethnicity and the racial differences of AMH cutoff value for the diagnosis of PCOS with possible interpretation.
Response 2:Thank you for this comment. The relevance of ethnicity and the racial differences of AMH cut-off value for the diagnosis of PCOS had been discussed in the introduction section, lines 37-40 and 60-63, and discussion section lines 207-210.
Point 3:Since the authors mention in the title ‘association with clinical symptoms’, there should more analysis of this data, in particular regarding the AMH levels in various phenotypes of PCOS.
Response 3:We studied the correlation of AMH with clinical symptoms of PCOS by Person’s correlation and chi-squared tests, and then logistic regression analysis was performed between AMH cut-off and PCOM, OA or age at menarche.
Point 4:I suggest to specify the evaluation of hirsutism with Ferriman-Gallwey score.
Response 4:Thank you for this comment. Yes, the evaluation of hirsutism was done with Ferriman-Gallwey score and this was clarified in the materials and methods.
Round 2
Reviewer 1 Report
The Authors modified the paper according to the Reviewers' suggestions so I have no further changes to suggest.